# State-of-the-Art in Sustainable Machining of Different Materials Using Nano Minimum Quality Lubrication (NMQL)

Avinash Kumar [1], Anuj Kumar Sharma [2] and Jitendra Kumar Katiyar [3,*]

1   Department of Mechanical Engineering, Indian Institute of Information Technology Design & Manufacturing Kancheepuram, Chennai 600127, Tamil Nadu, India
2   Centre for Advanced Studies, Dr. A P J Abdul Kalam Technical University, Lucknow 226021, Uttar Pradesh, India
3   Department of Mechanical Engineering, SRM Institute of Science & Technology, Kattankulathur, Chennai 603203, Tamil Nadu, India
*   Correspondence: jitendrv@srmist.edu.in; Tel.: +91-8090113301

**Abstract:** In the manufacturing industry, during machining, the conventional cutting fluid plays a vital role; however, extravagant use of cutting fluids due to its disposal affects the environment badly. Nowadays, due to these advantages of conventional cutting fluids, alternative methods of conventional cutting fluids or alternative methods are preferred. One of the most preferred methods may be the minimum quantity lubrication technique with conventional or nanoparticle-enriched cutting fluids. The present paper has a compilation of the investigations based on MQL application in different machining processes such as turning, milling, grinding, and drilling. The machining also involves hard-to-machine alloys. The paper discusses cryogenic MQL in brief and opens the domain for work in future. The purpose of this paper is to provide a quick reference for researchers working on the practical use of MQL lubricants with nanopowders dissolved and their application in machining for different materials.

**Keywords:** MQL; NMQL; nano cutting fluid; cryogenic; machining





## 1. Introduction

Traditionally, manufacturers use cutting fluids during any type of machining to reduce the cutting temperature force and friction that further reduce the production cost [1]. Traditional cutting fluids are of two types, namely neat cutting oils and water-soluble fluids [2]. Some of the advantages associated with the usage of cutting fluids during machining include (i) restricting the chemical diffusion, (ii) to take away the chips from the rake face, (iii) to decreases tool wear, (iv) reducing the power consumption, and (v) to guard the machined surface against corrosion.

Nevertheless, the performance of cutting fluids depends on parameters such as workpiece material, machining operation employed, and cutting parameters [3]. Additionally, several application techniques are employed in traditional cooling systems, such as spotlighting cooling by coolant flooding, mist, and high-pressure system cooling [4,5]. Despite various advantages of traditional coolants, there are also many disadvantages associated, i.e., adverse effects on tool, difficulty in achieving good surface finish, adverse effects such as skin allergies to the operators, high cost of coolants, difficulty to dispose, and their more significant impact on the environment [6,7].

The previously mentioned limitations of conventional cooling led to research on finding a cooling or lubricating strategy that uses eco-friendly cutting fluids. Dry machining is one of the most promising strategies developed, and its main objective is to save the environment by minimizing the usage of cutting fluids during machining. However, abandoning the cutting fluids during machining operation has disadvantages such as increased cutting temperature, friction between tool and work, poor chip removal, etc.

Though the advantage of dry machining includes low thermal shocks in cutting, it generates adhesive tool wear. That is why dry machining is not suitable for machining all materials. It is advantageous in machining materials such as aluminum, cast iron, and steel for drilling, reaming, sawing, tapping, and thread forming [8]. The aerospace materials such as Ni, Co-Cr, and Ti alloys that require high cutting speeds to process cannot be processed by dry machining [9].

It is mentioned that in some of the machining processes, the use of conventional cutting fluids or other lubrication systems is essential. Other alternatives such as solid lubricants, gaseous refrigeration, cryogenic refrigeration, or MQL system can be employed. A lot of experimental investigations have been carried out by researchers on the application of the MQL system in the machining of conventional and hard-to-machine alloys. However, more reviews on these investigations are needed for the researchers entering this field. Many research gaps are available for future research in the field of MQL machining, especially nano MQL. Therefore, this work focuses on compiling the previous investigations so that a comprehensive state-of-the-art can be presented with some research gaps on the application of MQL systems in machining different materials. The paper discusses the MQL in detail and its application in previous experimental investigations in other machining processes such as turning, milling, grinding, and drilling of hard alloys. Therefore, the paper has been sectioned into two parts, with Part-I focusing on experimental work carried out by several researchers with MQL-added nanoparticles and Part II covering the type of lubricants and the corresponding combination of nanoparticles.

## 2. Minimum Quality Lubrication

Minimum quantity lubrication (MQL) is an advancement to the traditional lubrication technique in machining i.e., the use of flood coolant during machining. MQL uses a biodegradable fluid whose minimum quantity gives a better lubrication effect as compared to high-quantity flood coolants. The disadvantages of flood coolants are messy shop floor, spillage, wet chips, adverse effects on the cutting tool, complicated lubrication set-up, skin allergies, and disposal issues. Figure 1 illustrates the setup of MQL in which MQL mist i.e., the aerosol is generated by mixing compressed air with cutting fluid. When the aerosol mist is sprayed while machining, it creates a lubricating coating layer at the tool-work interface which helps in reducing the friction generated and further aids to carry the heat generated through chips which are clean. Lacalle et al. [10] investigated the effect of spray-cutting fluids in high-speed milling. They obtained that the MQL flow penetrates the cutting zone and it acts in three different ways, that is, cooling tool and workpiece, lubricating and removing the chips effectively. The MQL strongly influences the cutting temperature over a wide range of speeds and lends itself to a lower cutting tool wear rate compared to completely dry machining [11]. But the use of cutting fluid having lower thermal conductivity, even with the help of the MQL system, cannot fully fill the need for green machining. Another technique that seems more appropriate is spraying the high thermal conductivity cutting fluid using MQL.

Saidur et al. [12] and Kakaç and Pramuanjaroenkij [13] in their work concluded that nanofluids significantly improve the heat transfer capability of conventional heat transfer fluids such as oil or water by mixing nanoparticles in these base fluids and concluded that the improved thermal conductivities of nanofluids are one of the driving factors for enhanced performance in different applications. A new class of cutting fluids can be synthesized by mixing metallic, non-metallic, ceramics or carbon nanoparticles in a conventional cutting fluid because, as compared with suspended mm or micro-sized particles, nanofluids show better rheological properties, stability, dramatically higher thermal conductivities, and no penalty in pressure drop [14]. The inclusion of nanoparticles of metal oxides in base fluid enhances the thermal conductivity of cutting fluids [15]. Chen and Ding [16] reviewed the research work regarding the enhancement in thermal conductivity of metallic, carbon, inorganic, and carbide materials. They found that thermal conductivity increases with the increase of particle volume concentration. Eastman et al. [17] found that the thermal

conductivity of 0.3% copper nanoparticles of ethylene glycol nanofluids is increased by up to 40% compared to base fluid. Liu et al. [18] noticed a 23.8% improvement in thermal conductivity at 0.1% volume fraction of copper particles in cutting fluid due to increased surface area. Moreover, Yoo et al. [19] argued that surface to volume ratio of nanoparticles is a significant factor that influences the thermal conductivity of nanofluids. Surface to volume ratio is increased with the addition of a smaller size of nanoparticles. Few researchers, such as Choi et al. [20] and Yang et al. [21], reported a very high enhancement in thermal conductivity of cutting fluids with the addition of MWCNT up to 150% and 200%, respectively as compared to the base fluid. Therefore, using nanofluids with MQL may be a better approach to achieving good machining results.

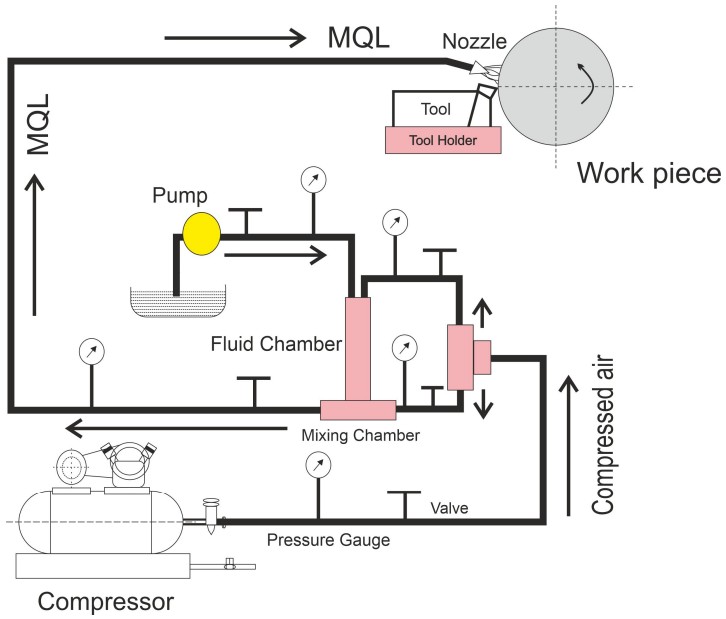

**Figure 1.** Schematic of MQL system.

Furthermore, Pareira et al. [22] discussed the sustainability analysis of lubricant oils for MQL-based machining on their tribe-rheological performance and observed an improvement in the tool life by up to 30% compared with commercial canola oil. Polvorosa et al. [23] effectively investigated tool wear on nickel alloys with different coolant pressures and recorded different insert nose wear patterns. Amigo et al. [24] combined high feed turning with cryogenic cooling on Haynes 263 and Inconel 718 superalloys, investigated the cutting forces under cryogenic cooling, and opened a path for combining two technologies.

Here it mentions that nanoparticles can be harmful to human being. However, few researchers have reported on non-toxic nature and applications of nano-particles in their investigations. Vallim et al. [25] have reported that nano encapsulated dimethoate reduces the toxicity of insecticides on zebrafish larvae and suggested that nanoencapsulation may be safer for non-target species by eliminating collateral effects and thus promoting sustainable agriculture. Stephen et al. [26] synthesized ZnO nanoparticles using the leaf extract of Averrhoa bilimbi Linn. The synthesized ZnO nanoparticles showed strong antioxidant potential as exhibited through 2,2-diphenyl-1-picryl-hydrazine-hydrate (DPPH) assay in a dose-dependent manner. Jaswal and Gupta [27] concluded that by the green formation of silver nanoparticles (AgNPs), the toxicity of silver nanoparticles could be reduced. Apart from the sustenance cutting fluid, the life cycle analysis (LCA) has a vital role in sustenance analysis. Khanna et al. [28] observed lower environmental impacts for MQL machining. They recorded 75% reduced tool life but with a higher cutting force and surface roughness compared to wet and cryogenic machining. Mia et al. [29] performed the LCA of cryogenic $LN_2$-assisted machining. It was found that $LN_2$ dual-jets were most effective in reducing

the specific energy consumption and temperature and improving surface quality. This can be attributed to the faster and more effective heat removal by $LN_2$ from the critical interfaces. It was found that LCA showed that there is a clear relation between cooling strategy and environmental aspects, such as resources, energy, human health, biodiversity, etc., that directly affect the machining performance of Ti-6Al-4V alloy. Agrawal et al. [30] investigated the machinability and sustainability of newly developed environment-friendly hybrid techniques for Ti-6Al-4V turning in terms of energy consumption, surface finish, and machining costs. They observed that the active energy consumed by machine tools is decreased by up to 1.8–3.6% using hybrid turning techniques compared to crycryogenic-assisted turning technique. An increase in energy efficiency values is observed at higher levels of input process parameters using all three turning techniques. Average surface roughness values are decreased by up to 32–42% using hybrid turning techniques compared to the cryogenic-assisted turning technique. Gupta et al. [31] evaluated the effectiveness of dry, liquid nitrogen ($LN_2$) and hybrid cryogenic and minimum quantity lubrication ($LN_2$ + MQL) conditions in terms of essential machinability indicators, for instance surface roughness, cutting forces, and temperature. They observed the superior cooling/lubrication effect under $LN_2$ + MQL conditions lowering the machining and environmental indices. The improvement in cycle time and productivity of $LN_2$ and $LN_2$ + MQL appeared to be 29.01% and 34.21% compared with dry turning. The sustainability assessment results also revealed that the lower cutting parameters under $LN_2$ + MQL produced the best results in achieving the overall sustainability index.

Chetan et al. [32] compared the techniques such as cryogenic cooling and cryogenic treatment with the $Al_2O_3$ nanoparticles-based MQL cooling method (nMQL) to turn the nickel-based Nimonic 90 alloy. They observed that the practical cushioning effect of liquid nitrogen as coolant improved the surface finish compared to other techniques. The cryogenic cooling reduced the defects such as material side flow, material redeposition, and microcavities over the machined surface. Overall, the cryogenic cooling environment is the best mode for machining the Nimonic 90 alloy. Sarikaya et al. [33] presented the machining and sustainability characteristics of minimum quantity lubrication (MQL), nanofluids-MQL, Ranque-Hilsch vortex tube MQL (RHVT + MQL), cryogenic-MQL as alternatives to flood cooling applications in the cutting of light-weight materials and concluded that MQL advancements could offer clear guidelines to implement hybrid cooling techniques to improve heat transfer, lubrication, and sustainable implementations. Shah et al. [34] found improved machining performance with lower power consumption with $LCO_2$, and it has higher impacts on ecology across 17 of the 18 impact categories of the LCA.

Astakhov and Woodhead [35] mention that designing the cutting tools and the cutting tool material should be selected especially for MQL. Moreover, the tool geometry and chip-breaking problems should be one of the prime concerns as if this problem is not resolved; it can quickly turn the whole MQL operation into a production nightmare. The designs of the aerosol-supply channels, tool geometry, tool materials, and design of the tool body (back tapers, reliefs, undercuts and supporting elements) should be optimized for MQL. The selection of cutting fluid in MQL depends on properties such as biodegradability, oxidation, and storage stability. There are several parameters involved in MQL-assisted machining such as type of cutting fluid, nozzle configuration, volumetric flow, or pressure to assess the tool life. A convenient way to predict the tool life, as developed by Marksberry and Jawahir [36] in dry machining using the MQL system, is given by the following equation.

$$T = T_R \frac{km}{f^{n1}d^{n2}} \left(\frac{V_R}{V}\right)^{\left(\frac{1}{n_c}\right)\left(\frac{1}{N_{NDM}}\right)}$$

where $T$ = tool life, $T_R$ = reference tool life for 1 min, $k$, $n_1$, $n_2$ = empirical constants, $m$ = constant depending on manufacturing process, $f$ = feed rate, $d$ = depth of cut, $V_R$ = reference cutting speed of 1 min tool life, $V$ = cutting speed, $n_c$ = coating effect factor, $N_{NDM}$ = contribution of the cooling system in the process.

## 3. Experimental Study

Several researchers have studied the usage of pure vegetable-based oils and mineral oils during different machining operations on various types of materials and observed that pure vegetable oils or mineral oils have lubricating properties such as low thermal conductivity, low wettability, high coefficient of friction, and low viscosity due to which researchers did not get promising results in their studies, and this made the researchers to extend their research on such type of cutting fluids which should have optimal lubricating and eco-friendly properties as well and came up with an idea called nanofluids. Nanofluids are the same vegetable or mineral oils used earlier with the dispersion of metallic nanoparticles with high lubricating properties such as high thermal conductivity, low coefficient of friction, and good chemical stability. Adding nanoparticles in the pure oils enhances the lubricating performance of pure oils. These nano-cutting fluids usage, along with MQL, improves the sustainability of machining operations by reducing the cutting forces, cutting temperature, tool wear, and surface roughness words have been proven by following researchers with their experimental studies.

### 3.1. Nanofluid-Assisted MQL Turning

Many researchers performed the turning operation on various materials as workpieces, and the summarized data is given in Table 1. Further, the following section compiles the investigations on turning operations on different materials.

**Table 1.** Summarized data of turning operation on various materials using NMQ.

| Ref. No. | Work/Tool Material | Base Fluid | Nano Particles/ Size in nm | Cutting Parameters | | | MQLParameters | | Effects (Reduction of Responses) |
|---|---|---|---|---|---|---|---|---|---|
| | | | | $V_c$ (m/min) | f (mm/rev) | d (mm) | Pressure (Bar) | Flow Rate (mL/Min) | |
| [37] | AISI1040steelØ70 mm and 300 mm L/ Cemented carbide | Vegetable oil + water emulsion | $Al_2O_3$/45 | 96.7 | 0.1 | 1 | 4 | 50 | 1% conc.—cutting force, flank wear and surface roughness. |
| [38] | AISI-1040 steel Ø70 mm and 300 mm L/Carbide | Vegetable oil | $TiO_2$/ <100 | 96.7 | 0.1 | 1 | 5 | 50 | 1% conc. $TiO_2$-tool wear, surface roughness and cutting force by 50, 30, and 30% compared to other environments |
| [39] | AISI D2 steel Ø40 mm and 180 mm L/Tungsten carbide | SAE20W40 mineral oil | MWCNT/ <100 | 51–123 | 0.1–0.2 | - | 5–7 | - | 0.2 wt.%—surface roughness and cutting temperature |
| [40] | AISI 304 stainless steelØ70 mm and 300 L- | Water emulsion | MWCNT/ <100 | 40 | 0.08 | 0.6 | 4 | 0.83 | 1% conc.—cutting temperature and surface roughness at lower value of cutting parameters |
| [41] | Austempered ductile iron/Carbide | Ecolubric E200 vegetable oil | $Al_2O_3$/ 20 | 120 | 0.2 | 0.5 | 5 | 40 | 4 vol.% con.—tool wear |
| [42] | H 11 steel Ø20 mm and 100 mm L/Carbide | Ethylene glycol | Cu/ 50 | 209 | 0.1 | 1 | 3 | 7 | 0.2 wt.% con.—surface roughness and flank wear by 40% and 60% |
| [43] | AISI-H13 steelØ50 mm and 250 mm L/Tungsten carbide | Coconut oil with additives + Mineral oil | $CaF_2$ and $MoS_2$/ <100 | 90 | 0.28 | 0.5 | 5 | 0.58 | 0.3 wt.%. con. $MoS_2$ (HN-GCF) 1:16-co-efficient of friction, surface roughness and tool wear |
| [44] | AISI-4340 steel Ø24 mm and 100 mm L/Tungsten-coated carbide | Ethylene glycol | MWCNT/ <70 | 75 | 0.04 | 1 | 5 | 2.33 | 0.2% conc.—surface roughness and cutting force |
| [45] | AISI 4340 steel Ø50 mm and 700 mm L/Cermet | Radiator coolant | $Al_2O_3$/ <50 | 80 | 0.05 | 0.4 | 7 | 2.5 | tool wear, cutting force and serrations on chips |
| [45] | EN24 steelØ20 mm and 100 mm L/Carbide | Ethylene glycol | Cu/ 20–50 | 157 | 0.2 | 0.2 | 3 | 10 | 0.4 wt.% con.—surface roughness and flank wear |
| [46] | AISI-D3 steel Ø20 mm and 145 mm L/Tungsten carbide | Ethylene glycol | Graphene/ <100 | 31.4–94.2 | 0.03–0.13 | 0.75–1.5 | 2 | 8 | 0.8 wt.% con.—surface roughness and cutting temperature to 50% compared to other |
| [47] | 90CrSi steel Ø40 mm/Tungsten carbide | Soya bean oil + water emulsion | $Al_2O_3$ and $MoS_2$/ 30 | 81.7–119.4 | 0.1 | 0.15 | - | - | $Al_2O_3$—thrust force and surface roughness whereas MoS2-Cutting and feed forces |

**Table 1.** *Cont.*

| Ref. No. | Work/Tool Material | Base Fluid | Nano Particles/ Size in nm | Cutting Parameters | | | MQLParameters | | Effects (Reduction of Responses) |
|---|---|---|---|---|---|---|---|---|---|
| | | | | $V_c$ (m/min) | f (mm/rev) | d (mm) | Pressure (Bar) | Flow Rate (mL/Min) | |
| [48] | AISI-304 steel/Cemented carbide | Vegetable oil + distilled water + detergent | $Al_2O_3$-MWCNT/ 45 | 60–120 | 0.08–0.16 | 0.6–1.2 | 4 | 2.5 | 1.25% conc. hybrid Al-MWCNT (90:10)—cutting forces and surface roughness |
| [49] | AISI-52100 steel/Carbide | Blaser cutting oil in DI | $Al_2O_3$, $Al_2O_3$-Graphene/ 45 and 10–16 | 90 | 0.1 | 0.5 | 6 | 5 | 0.75% conc. Al-graphene (85:15)—cutting power, machine tool power, surface roughness and specific energy consumption |
| [50] | AISI-1080 Steel, Ø40 mm and 300 mm L/Ti-AlN | Coconut oil | $CuO$-$Al_2O_3$/ 50–100 and 30 | 180 | 0.1 | 0.5 | - | - | $CuO$-$Al_2O_3$ (50:50)—surface roughness |
| [51] | Inconel-600 alloyØ20 mm and 100 mm L/Carbide | vegetable oil-coolube 2210EP | $Al_2O_3$/ <100 | 40–60 | 0.08–0.16 | 0.4–1.2 | 5 | 1.66 | 6 vol.% con.—tool wear, cutting force, temperature and surface roughness compared to dry and pure MQL |
| [52] | Nicrofer-C263 Ø42 mm and 250 mmL/Cemented tungsten carbide | Max Mist ST2020 | $Al_2O_3$/<100 | 36–54.75 | 0.09–0.12 | 0.75–0.9 | 5 | 1 | 1 vol.% con.—surface roughness, cutting force and cutting temperature compared to dry and pure MQL |
| [53] | Nimonic90 Ø60 mm and 300 mm L/Tungsten Carbide | Water | $Al_2O_3$ and Ag/ 40 and 10 | 60 | 0.12 | 0.5 | 4 | 1–4.16 | $Al_2O_3$ 125 mL/h—tool wear, chip thickness and friction co-efficient |
| [54] | Inocel-617 Ø32 mm and 300 mm L/PVD (AlTiN) | Coconut oil | $Al_2O_3$/ <100 | 40–100 | 0.14–0.2 | 0.5 | 4 | 20 | 0.25% con.—tool wear, cutting force surface roughness and serrations on chips |
| [55] | Inconel-718 (ASTMSB 637)/Tungsten carbide | Ecolubric E200 vegetable oil | $Al_2O_3$ and MWCNT/ 20 and 13–20 | 40–60 | 0.2–0.4 | 0.2 | 5 | 0.66 | 4% con. MWCNT—cutting force and cutting temperature |
| [56] | Inconel-625 Ø70 mm and 300 mm L/PVD TiNcementedcabide | Plantocut 10 SR | h-BN/ 70 | 40 | 0.075 | 0.8 | 8 | 0.83 | 1%vol. con.—tool wear, cutting temperature and surface roughness |
| [57] | Inconel-800 Ø50 mm and 120 mm L/Cubicboron nitride | Sunflower oil | $Al_2O_3$, $MoS_2$ and Graphite/40 | 200–300 | 0.1–0.2 | 0.25–0.75 | 5 | 0.5 | 3 wt.% con. graphite-cutting forces, tool wear and surface roughness compared to others |
| [58] | Hastelloy-X/PVD (TiAlN) and CVD ($Al_2O_3$) | Coconut oil | h-BN/ 30-1 00 | 40–100 | 0.14–0.2 | 0.5 | 4 | 0.125 | 0.25 vol.% con with PVD-coated tool-tool wear, cutting force and surface roughness |

Table 1. *Cont.*

| Ref. No. | Work/Tool Material | Base Fluid | Nano Particles/ Size in nm | Cutting Parameters | | | MQLParameters | | Effects (Reduction of Responses) |
|---|---|---|---|---|---|---|---|---|---|
| | | | | $V_c$ (m/min) | f (mm/rev) | d (mm) | Pressure (Bar) | Flow Rate (mL/Min) | |
| [59] | Titanium (grade-2) Ø52 mm and 150 mm L/Cubic boron nitride | Vegetable oil | $Al_2O_3$, $MoS_2$ and Graphite/ 40 | 215 | 0.1 | 1 | 5 | 0.5 | Graphite—cutting forces, tool wear and surface roughness compared to $Al_2O_3$ and MoS2 |
| [59] | Ti-6Al-4V (Titanium grade5) Ø30 mm and150 mm L/CNMG 12408 | soya bean oil | Graphene/ <100 | 100–200 | 0.1–0.2 | 0.2–0.4 | - | 16.66 | Graphene—flank wear, surface roughness and cutting temperature compared to dry and pure MQL environments |
| [64] | Ti-6 Al-4V(UNSR56400)/Carbide | Ecolubric E200 | MWCNT/ 13–20 | 120–220 | 0.1–0.2 | 0.2 | 5 | 0.66 | 2% conc.—power consumption and tool wear by 11.5% and 45% compared to base fluid MQL |
| [65] | Titanium grade2 Ø16 mm and 130 mm L/ tungsten carbide | Ethylene glycol | Ag/ 20 | 85.4 | 0.03 | 1.5 | 3 | 10 | 0.4 wt.% con.—tool wear, surface roughness and cutting temperature |
| [66] | Ti-6 Al-4V Ø30 mm and 200 mm L/carbide | Jojoba oil and LRT 30 | $MoS_2$/ 80–100 | 80 | 0.16 | 4 | 6 | 1 | 0.1 wt.% of MoS2 in jojoba oil—tool wear, cutting force, surface roughness |
| [67] | Ti-6Al-4V Ø80 mm and 300 mm L/laser textured carbide | Sun flower oil + deionized water | $Al_2O_3$/ 40 | 60–120 | 0.1–0.2 | 1 | 4–7 | 1.16–1.66 | Sun flower oil + deionized water in the ratio 1:10 with MQL improved the machining performance compared to dry and nano MQL |
| [69] | Ti-6Al-4V (Ø50 mm and 250 mm L)/CNGA-120408 T01020-WG | Blaser-distilled water base fluid | CuO-MWCNT/ 5–10 and 10–30 | 80–120 | 0.08–0.12 | 1 | 4.5 | 2.5 | 24% con. CuO- MWCNT (90:10)—tool wear, surface roughness, cutting temperature and power |
| [70] | Haynes-25 (L = 100 mm)/carbide | Blasar vegetable oil in water | $Al_2O_3$- Graphene/ 45 | 30–60 | 0.08–0.16 | 1 | 6 | 5 | (85:15) Al-graphene—cutting energy, carbon emission and cost per part. |

### 3.1.1. Turning on Steel

Sharma et al. [37] investigated the sustainable turning of AISI-1040 steel using vegetable oil-based $TiO_2$ and $Al_2O_3$ nanofluids under MQL on cutting force, tool wear, and surface roughness. The results showed that $TiO_2$ nanofluid gave a significant reduction in cutting force, whereas $Al_2O_3$ nanofluid was best in reducing tool wear and surface roughness. The tool performance was greatly improved, as investigated by Sharma et al. [38,39], with mineral oil-based MWCNT nanofluid under MQL in turning AISI-D2 steel due to a reduction in cutting temperature and surface roughness. The cutting parameters and nanoparticles concentration i.e., lower values of $V_c$, f, and d and 0.1 wt.% MWCNT water-soluble-based nanofluid played a significant role in achieving superior lubricating performance in the turning study by Singh et al. [40] on AISI-304 stainless steel under MQL. A 4% (Vol.) vegetable oil-based $Al_2O_3$ nanofluid greatly reduced the tool wear during the turning of austempered ductile iron (ADI), as reported by Eltaggaz et al. [41]. Ethylene glycol-based Cu nanofluid during MQL turning of deep hardened H11 steel reduced the surface roughness and flank wear by 40% and 60%, respectively as compared to dry and pure MQL cutting according to the investigation by Ganesan et al. [42]. $MoS_2$ suspended in a hybrid mix of green cutting fluids, and mineral oil (HN-GCF) is observed to be the best with reduced coefficient of friction and surface roughness while turning H13 steel as per the study carried out by Gajrani et al. [43]. Furthermore, Patole et al. [44] reported the lubrication performance of MWCNT nanofluids during the MQL turning of AISI-4340 steel by analyzing the cutting forces and surface roughness. A further advancement on AISI-4340 steel, i.e., finding the cutting force components such as feed, cutting, and thrust forces was carried out by Das et al. [45] and evaluated the performance of nanoparticles with MQL using $Al_2O_3$ nanoparticles. EN24 is high-strength steel used in high-strength shafts and gears, and its machining performance was improved significantly with MQL mist mixed with Cu nanofluid as studied by Babu et al. [46]. The ethylene glycol-based graphene nanofluid greatly influenced the machining performance of AISI-D3 steel by reducing the tool wear and surface roughness by 85 % and 44% and cutting temperature by 53%, respectively as compared to dry and oil-based MQL environments, Babu et al. [47]. Duc et al. [48] claimed that soya bean oil-based $Al_2O_3$ nanofluid with MQL showed the best lubricating performance compared to $MoS_2$ nanofluid by reducing the thrust forces generated during hard finishing of 90CrSi steel (60-62 HRC) and surface roughness significantly. The muted response measurement study by Sharma et al. [49], i.e., cutting forces involved in MQL turning of AISI-304 stainless steel, was found during his research on the sustainable turning of AISI-304 stainless steel using Al-MWCNT hybrid nanofluid. The study showed that hybrid 1.25% concentrated Al-MWNCT nanofluid enhanced the machining performance of AISI-304 steel by greatly reducing the cutting force compared to using $Al_2O_3$ nanofluid alone. Hybrid nanofluids provide better lubricating performance compared to unit nanofluids by which a superior surface finish can be obtained. This statement was demonstrated by Khan et al. [50] by conducting turning experiments on AISI-52100 steel using $Al_2O_3$-Graphene hybrid nanofluid under an MQL environment where he found that cutting power, machine tool power, surface roughness, tool wear, and cost per part are better as compared with $Al_2O_3$ nanofluid. This also resulted in a reduction of specific energy consumption. According to Kumar et al. [51], MQL with coconut oil-based (50:50) $CuO-Al_2O_3$ nanofluid combination gave superior work quality by a significant reduction in surface roughness while turning AISI-1080 steel.

### 3.1.2. Turning on Nickel Alloys

$Al_2O_3$ nanofluid of 6% (Vol.) concentration with MQL enhanced the machining performances of Inconel-600, a good oxidation-resistant material at high temperatures by reducing the tool wear, surface roughness, cutting temperature, and cutting forces as per the study performed by Vasu et al. [52]. Bose et al. [53] reported that 1 g of $Al_2O_3$ nanopowder in 100 mL of vegetable oil greatly reduced the surface roughness, cutting force, and cutting temperature during MQL machining Nicrofer-C-263. Turning Nimonic-90 alloy, a highly

heat-resistant material used for turbine blades under MQL with sunflower oil and $Al_2O_3$ nanofluid at a medium-level flow rate, i.e., nanofluid., 125 mL/h, enhanced its machining performance significantly compared to Ag nano flu, as claimed by Chetan et al. [54]. Mathew et al. [55] studied the machinability of Inconel-617 by performing a series of turning experiments under MQL with PVD-coated AlTiN tool and found that coconut oil with $Al_2O_3$ nanofluid improved the machinability of Inconel-617 by reducing the surface roughness, flank wear, and cutting force. The comparison effect of $Al_2O_3$ and MWCNT nanofluids on turning Inconel718 was studied by Hegab et al. [56] who reported that 4 vol.% MWCNT reduced the tool wear by enhancing the heat dissipation significantly as compared to $Al_2O_3$. Yıldırım et al. [57] investigated the use of highly conductive nanofluid, i.e., ester-based h-BN nano cutting fluid during MQL turning of Inconel625 and found that machining performance was greatly improved at 1 vol.% h-BN, $V_c$ = 40 m/min and $f$ = 0.075 mm/rev. Graphene nanofluid showed the best lubricating performance as compared to $MoS_2$ and $Al_2O_3$ nanofluids by reducing the cutting forces, tool wear, and surface roughness significantly as investigated by Gupta et al. [58] on Inconel-800. According to Venkatesan et al. [59], adhesive and abrasive wear of the PVD (AlTiN)-coated tool is better than CVD ($Al_2O_3$) in turning Hastelloy-x super alloy with h-BN nanofluid.

Further, Valdivielso et al. [60] developed an indirect method for seeking standard features in the group of those tools with the best performance on machining Inconel 718. The technique is executed into five stages, which provide knowledge based on the distilling of results, identifying carbide grades, chip breakers shapes, and other features for having the best tool performance. All surface integrity effects are checked for the best solution. This new point of view is the only way to improve difficult-to-cut alloy machining, reaching technical conclusions with industrial interest. The results show the method applied on Inconel 718 turning, resulting in a carbide grade with 10% cobalt, submicron grain size (0.5–0.8 μm), and hardness around 1760 HV, coating TiAlN monolayer with 3.5 μm thickness, chip breaker giving 19° of rake angle that becomes 13° real one after the insert is clamped on the tool holder.

Furthermore, Jadhav and Mohanty [61] attempted MQL and cryogenic machining process during turning the Nimonic C-263 workpiece to achieve an ideal machining environment. The study revealed that the cryogenic machining strategy is adequately proficient over MQL machining to deliver energy professional and gratifying work environment for the tool engineers by reducing the machining cost and improving their work efficiency. Marques et al. [62] explored an approach to fostering sustainability in metal cutting lubrication to supply solid lubricant at the minimum quantity and to study the effect of applying solid lubricant (molybdenum disulfide and graphite) mixed with oil during the turning of Inconel 718 using cemented carbide tools. Results show that a minimum quantity solid lubricant consisting of molybdenum disulfide and oil mixture performed better. Therefore, it may be a cost-effective and environmental-friendly lubrication technique compared to flood coolant and sprayed oil with or without graphite to retard all types of destructive processes and to improve the machinability characteristics of Inconel 718.

### 3.1.3. Turning on Titanium Alloys

Gupta et al. [63] investigated the effect of$Al_2O_3$, $MoS_2$, and graphite nanofluids on the machining of widely used titanium grade-2 alloy and found that graphite nanofluid enhances the machining performance by reducing the cutting force, temperature, tool wear, and surface roughness significantly as compared to the other two. Tool life was improved with a reduction in flank wear while investigating Ti-6Al-4Valloy by Katta et al. [64], who used soya bean oil-based graphene nanofluid. Ecolubric E200-based MWCNT nanofluid reduced the tool wear and power consumption to 45% and 11.5% while machiningTi-6Al-4V under MQL by Hegab et al. [65]. A reduction in flank wear, surface roughness by 60% and 26% and cutting temperature and forces by 34% and 18%, respectively was observed by Anandan et al. [66] while using silver nanofluid during turning titanium grade-2 alloy as compared to flood lubrication. Gaurav et al. [67] evaluated the lubrication

performance of MoS$_2$ nanoparticles in two different types of oils, i.e., jojoba (vegetable) and LRT30 (mineral oil) during MQL turning of Ti-6Al-4V and the results showed that jojoba-based MoS$_2$ nanofluid with MQL enhanced the machining performance significantly as compared to the later one. Nanofluid with MQL is not suitable for machining Ti-6Al-4V in combination with a laser-textured tool due to the agglomeration of nanoparticles on the textured face of the tool, as demonstrated by Mishra et al. [68]. These were investigated in the sustainable turning of Ti-6Al-4V under MQL using Al nanofluid with a circular patterned laser textured tool. According to Jamil et al. [69], blaser-distilled water-based CuO-MWCNT hybrid nano-cutting fluid with MQL achieved clean machining of Ti-6Al-4V as compared to CO$_2$ snow with a significant reduction in tool wear, power, surface roughness, and cutting temperature.

### 3.1.4. Turning on Cobalt Alloys

The cutting power, energy consumption, carbon emission, and cost per part were observed to be reduced with the usage of alumina-graphene hybrid nanofluid due to the rolling and protection mechanism of nanoparticles b/w the tool work interface during the investigations by Khan et al. [70] on sustainable MQL turning of Haynes-25, a Co super alloy, which is used in high-temperature applications such as jet engine parts. This presents a strong possibility of using Haynes 25 alloy for industrial applications.

### 3.2. Nanofluid-Assisted MQL Milling

Many researchers performed the milling operation on various materials as work-pieces, and the summarized data is given in Table 2. Further, the section compiles the investigations on milling operations carried out on different materials.

**Table 2.** Summarized data of milling operation on various materials using NMQL.

| Ref. No. | Work/Tool Material | Base Fluid | Nano Particles/ Size in nm | Cutting Parameters | | | MQL Parameters | | Effects (Reduction of Responses) |
|---|---|---|---|---|---|---|---|---|---|
| | | | | $V_c$ (m/min) | f (mm/rev) | d (mm) | Pressure (bar) | Flow Rate (mL/Min) | |
| [71] | AISI-1045 steel (203.2,127 and 203.2)/TIAlN-coated carbide insert (Ø25 mm) | UNIST Coolube 2210 (vegetable oil) | Graphene platelets/ 10 thick | 274–353 | 0.5–0.7 | A = 1 and R = 0.6 | 5.5 | 1.5 | 0.1 wt. % con.—tool wear and friction co-efficient |
| [72] | AISI-1050 and AISI-P21 steel (100 × 100 × 80 mm³)/Carbide TiN+TiAlN (Ø20 mm) | Vegetable cutting oil | MWCNT/ 24–40 | 157 | 0.1 | A = 0.8 and R = 20 | 4 | 1 | 0.5 wt. % con.—surface roughness and tool wear compared to dry and wet milling |
| [73] | AISI-420martensitic steel (400 × 250 × 4 mm³)/Tungsten carbide end mill (Ø32 mm) | Eraoil KT/2000 | MoS$_2$/<60 | 99 | 0.18 | 0.5 | 5 | 0.33–0.66 | 1% wt. con.—tool wear and surface roughness compared to other cutting environments |
| [74] | AISI-430 ferritic steel (400 × 250 × 6 mm³)/Tungsten carbide end mill and TiN-coated Tungsten carbide tool (Ø32 mm) | Eraoil KT/2000 | Graphene/<100 | 100 | 0.18 | 0.5 | 5 | 0.33–0.66 | TiN-coated tool and 0.5 wt.% con.—cutting temperature and burs on surface of work |
| [75] | AISI-304 stainless steel (210 × 105 × 110 mm³)/Coated Carbide (Ø35 mm) | Polyethylene glycol 300 (PEG300) | GO nano sheets, SiO$_2$ and GO-SiO$_2$/ 5–10 thick and 20–30 dia. | 100 | 0.12 | A = 1 and R = 5 | 3.5 | 0.25 | GO/SiO2(0.02:0.50) con.—achieved significant tribological characteristics and milling performance compared to all other |
| [76] | AISI-304 austenitic stainless steel (210 × 105 × 110 mm³)/ Coated Carbide (Ø35 mm) | LB-2000 vegetable oil | Graphene/ 5–10 and ≤10 µm. dia. | 100 | 0.1 | A = 1 and R = 5 | 3 | 0.166 | 0.06 wt.% con. at -6 kv.—tool wear and surface roughness effectively compared to base EMQL |
| [77] | AISI-304 steel (200 × 150 × 50 mm³)/coated (Al,TiN) carbide insert | Soya bean oil | MWCNT/ 15-Oct | 100–160 | 0.075–0.15 | 0.3–0.6 | 6 | 1.25 | 1% wt. con.—surface roughness significantly compared to dry, flood, and pure MQL |
| [78] | AISI-4340 steel (250 mm × 100 mm × 20 mm)/cutter with Tungsten carbide insert (Ø16 mm) | Servocut -S oil + water | Boric acid, Graphite and Boric acid/ Graphite/<100 | 251–376 | 0.1–0.13 | 0.5 | 5 | 2.5 | 10 wt.% con.—cutting forces and surface roughness compared to others |

**Table 2.** *Cont.*

| Ref. No. | Work/Tool Material | Base Fluid | Nano Particles/ Size in nm | Cutting Parameters | | | MQL Parameters | | Effects (Reduction of Responses) |
|---|---|---|---|---|---|---|---|---|---|
| | | | | $V_c$ (m/min) | f (mm/rev) | d (mm) | Pressure (bar) | Flow Rate (mL/Min) | |
| [79] | AISI-O2 Steel (150 × 80 × 80 mm³)/Carbide (Ø25 mm) | Ethylene glycol | h-BN/<100 | 100 | 0.05 | A = 0.5 and R = 15 | 5 | 0.83 | 2 wt.% con.—tool wear, surface roughness, and cutting force |
| [80] | SKD 11 tool steel (90 × 48 × 50 mm³)/submicron carbide insert (Ø50 mm) | Emulsion-based cutting fluid | $MoS_2$/30 | 90–110 | 0.012 | 0.12 | 6 | 0.5 | 0.5 wt.% con.—surface roughness |
| [81] | SKH-9 Steel/double edge micro end mill(Ø300μm) | Oil | MWCNT and Graphene/ 12 and 60 | 37.69–56.54 | 0.002–0.004 | 0.1 | 1–3 | 0.25–0.58 | 1 wt.% con.—micro milling force, temperature and tool wear compared to other techniques employed |
| [82] | EN-GSJ 700–02 cast iron (70 × 160 × 40 mm³)/coated carbide cutting inserts (Ø32 mm) | ERALUBETM BIO CF 350 | $MoS_2$/90 | 300 | 0.2 | A = 1 and R = 10 | 3–5 | 2.6–5.16 | 0.5 wt.% con.—surface roughness, traces of abrasive and adhesive wear of tool |
| [83] | AA6061-T6 (50 × 50 × 200 mm³)/HSS tool (Ø10 mm) | ECOCUT SSN 322 mineral oil | $SiO_2$/ 15 | 157 | 0.02 | 5 | 200 | 2 | 0.2 wt.% con.—cutting force, specific energy and power during machining |
| [84] | AA6061-T6 alloy/Tungsten cobalt (6%) insert | De-ionized water | $TiO_2$/ 40 | 5200–5600 rpm | 0.09 | 2.25 | 6 | 0.65 | 2.5 wt.% con.—Higher adhesion and edge chipping |
| [85] | AA6061-T6 (100 × 100 × 20 mm³)/Coated tungsten carbide | Ethylene Glycol | $TiO_2$- ZnO/ 21 and 10–30 | 3000–6800 rpm | - | 0.35–1.3 | - | 0.6–2.4 | $TiO_2$- ZnO (80:20) 0.1 vol.%—tool wear and surface roughness compared to dry and pure MQL |
| [86] | AA7075-T6 alloy (152 × 103 × 80 mm³)/High speed steel (Ø10 mm) | ethylene glycol | Ag and Borax/ <100 | 64–135 | 0.029–0.171 | - | 5 | 0.83 | Reduced the surface roughness significantly but failed in reducing cutting force |
| [87] | Inconel-690/Uncoated carbide tool (6 mm) | Palm oil | $Al_2O_3$/ 30 | 140 | 0.2 | 1 | 8 | 2 | 2.5 wt.% con.—specific cutting energy, surface roughness, cutting temperature and tool wear compared to other medium |
| [88] | Inconel-X750 alloy (100 × 150 × 17.3 mm³)/Coated carbide TiAlN | Vegetable oil | h-BN, $MoS_2$ and graphite/ 80 | 45 | 0.10 | 0.5 | 8 | 0.83 | 0.5 wt.% con.—cutting force, temperature, and surface roughness |

**Table 2.** *Cont.*

| Ref. No. | Work/Tool Material | Base Fluid | Nano Particles/ Size in nm | Cutting Parameters | | | MQL Parameters | | Effects (Reduction of Responses) |
|---|---|---|---|---|---|---|---|---|---|
| | | | | $V_c$ (m/min) | f (mm/rev) | d (mm) | Pressure (bar) | Flow Rate (mL/Min) | |
| [89] | Nickel alloy X-750/uncoated SiAlON CC cutting tool | Belgin oil cuttex syn.5 | h-BN/ 65–75 | 500–700 rpm | 0.025–0.075 | A = 0.5 and R = 15 | 8 | 0.83 | 0.5 vol.% con.—flank wear, surface roughness, cutting force, and temperature |
| [90] | HastelloyC276 ($150 \times 100 \times 15$ mm$^3$)/coated carbide inserts (Ø32 mm) | Vegetable oil | $Al_2O_3$/ 18 | 60–90 | 0.1–0.2 | - | 8 | 1.6 | 1 wt.% con.—significant improvement in surface roughness and tool wear |
| [91] | Ti-6Al-4V ($30 \times 30 \times 5$ mm$^3$)/tungsten carbide end mill (Ø500 µm) | Neo-01 vegetable oil | Diamond/ | 70.6 | 0.005 | 0.1 | 1.5 | 0.16 | 0.1 wt.% con.—milling forces, surface roughness, tool wear, and co-efficient of friction |
| [92] | Titanium TC4 alloy ($80 \times 30 \times 15$ mm$^3$)/coated carbide end mill (Ø6 mm) | LB2000 vegetable oil-based cutting fluid | Graphene/ 5 | 15 | 0.0.16 | 0.1 | 6 | 1 | 0.1 wt.% con.—surface roughness, milling forces, temperature, and tool wear |
| [93] | Ti-6Al-4V ($40 \times 30 \times 30$ mm$^3$) | Cotton seed oil | $Al_2O_3$, $MoS_2$, $SiO_2$, CNTs, SiC and graphite/ 70 | 1200 rpm | 0.41 | A = 0.25 and R = 10 | 4 | 1,41 | 1.5 wt.% spherical shaped nanoparticles $Al_2O_3$ and $SiO_2$ had improved the lubricating performance of base fluid compared to other nano fluids. |
| [94] | Ti-6Al-4V ($50 \times 50 \times 100$ mm$^3$)/Coated cemented carbide (Ø10 mm) | Fatty acid ester | h-BN/ 80–100 | 56–73 | 0.01–0.059 | 0.68–2.31 | 3 | 0.31–0.66 | 24.75%. con.—cutting forces and surface roughness compared to pure MQL |
| [95] | Ti-6Al-4V/Carbide milling tool (Ø10 mm) | Blaso cut oil emulsion | $Al_2O_3$-MWCNT/ 30 | 67.5–130 | 0.012–0.024 | A = 0.25–0.45 and R = 1.6–3.6 | 4 | 2 | 1 vol.% $Al_2O_3$-MWCNT (90:10) con.—surface roughness, energy consumption and enhanced the metal removal rate |

### 3.2.1. Milling on Steel

The wettability of vegetable oil was found to be improved with the addition of graphene nanoplatelets (1 μm dia.; 0.1 wt.%), due to which significant reduction in tool wear and friction co-efficient was observed while ball milling AISI-1045 material under MQL by Park et al. [71]. Huang et al. [72] clarified that MWCNT nanoparticles gives immense lubrication performance of vegetable oil due to which it achieves the best milling performance of AISI-1050 and AISI-P21 steels at high depth of cuts under MQL compared to flood coolant systems. Uysal et al. [73] proved that the machining performance of hardened AISI-420 martensitic steel could be enhanced by using 1%wt. concentrated vegetable oil-based $MoS_2$ nanofluid under MQL at 40 mL/h flow rate. Further, Uysal et al. [74] investigated AISI-430 ferritic stainless steel milling performance and machining quality, i.e., reduction of burrs, and found that they can be enhanced by using TiN-coated WC tool along with 0.5 wt.% vegetable oil-based graphene nano-cutting fluid. A significant enhancement of tool life and surface roughness is observed with 0.02:0.5 concentrated mineral-based $GO/SiO_2$ hybrid nanofluid with MQL, as reported by Lv et al. [75]. Electrostatic MQL (EMQL) is a synergetic technique of electrostatic spraying and MQL which creates a fine mist of (1–20 mL/h) lubricant than MQL was studied by Lv et al. [76] through conducting EMQL milling experiments on AISI-304 steel using vegetable oil-based graphene nanofluid. Here the graphene nanofluid showed the best tribological characteristics with improved machining performance and tool life under EMQL. Considering the economic point of view, Singh et al. [77] evaluated the machining performance of AISI-304 steel using vegetable-based MWCNT with an MQL system and found that 1 wt.% MWCNT reduces the flank wear mechanisms involved, such as adhesion, and abrasion, and surface roughness significantly at a cutting speed of 160 m/min, feed as 0.15 mm/tooth, and a depth of cut 0.3 mm. Muaz et al. [78] investigated the performance of boric acid, graphite, and their mixture during the milling of AISI-4340 steel and found that 10 wt.% boric acid concentrated servo cuts which are very cheap compared to the remaining, achieves significant machining performance by reducing the cutting forces and surface roughness. Kursuncu et al. [79] demonstrated that ethylene glycol-based h-BN nanofluid with MQL enhances the machining performance and tool life while milling $AISI-O_2$ cold-worked steel which is very hard and used in the manufacturing of cutting tools. Dong et al. [80] employed the technique of MQCL that comprises both MQL and Ranque-Hilsch vortex tube principles in machining difficult-to-cut materials, i.e., SKD 11 tool steel and found that MQCL with 0.5 wt.% $MoS_2$ nano fluid achieves the best surface quality with reduced burn marks. According to Huang et al. [81], graphene nanofluid under ultrasonic atomisation MQL is a superior lubrication technique that significantly reduces micro milling force, temperature, and tool wear while milling hardened SKH-9 high-speed steel.

### 3.2.2. Milling on Cast Iron

The reports by Çelik al. [82] on milling the EN-GSJ 700–02 ductile cast iron under MQL using 0.5 wt.% $MoS_2$ nanofluid at 5 bar pressure and 160 mL/h flow rate had given the most minor surface roughness with reduced abrasive and adhesive tool wear compared to dry and flood environments.

### 3.2.3. Milling on Aluminum Alloys

Sarhan et al. [83] show that milling AA6061-T6, a precipitation-hardened alloy used in frames of bicycles, fire rescue ladders, etc., under MQL using $SiO_2$ nanofluid reduces the cutting force, specific energy, and power during machining. The flank wear patterns, mechanisms, and the effect of cutting parameters on the tool and non-deterministic component of the sustainability index for the milling process with MQL in machining AA6061-T6 alloy were found by Najiha et al. [84]. The authors reported that primary tool wear mechanisms are micro abrasion, micro attrition and higher adhesion and edge chipping are observed with an increase in feed rate and depth of cut, which was significantly reduced with the optimal concentration of $TiO_2$ nanofluid i.e., 2.5 wt.%. Sahid et al. [85] investigated material

removal rate and surface roughness and proved that using $TiO_2$-ZnO hybrid nanofluid during milling of AA6061-T6 enhances the machining performance with a significant reduction in surface roughness. Cetin et al. [86] added borax in ethylene glycol nano-cutting fluid and showed the best lubricating performance on AA7075-T6 alloy under MQL with a significant reduction in surface roughness and cutting forces compared to Ag-added ethylene glycol nano-cutting fluid under similar conditions.

### 3.2.4. Milling on Nickel Alloys

The influence of palm oil-based $Al_2O_3$ nanofluid on the milling performance of Inconel 690 alloy by Sen et al. [87] found that 2.5 wt.% $Al_2O_3$ nanofluid had shown the best tribological performance by reducing the specific cutting energy, surface roughness, cutting temperature, and tool wear. The optimal milling performance of Inconel X-750 was achieved using vegetable oil-based 0.5 wt.% h-BN nano cutting fluid under MQL during the investigations carried out by Şirin et al. [88] with a significant reduction in cutting force, cutting temperature, and surface roughness compared to MoS2 and graphite nanofluids. Günan et al. [89] also supported the above investigations by further research on milling Inconel X-750 using h-BN nanofluid under MQL and clarified that significant reduction in flank wear and minimal difference in microhardness of work is possible with h-BN nano cutting fluid with MQL. According to Şirin et al. [90], vegetable oil-based 1 wt.%$Al_2O_3$ nanofluid with MQL improves the milling performance of Hastelloy-C276, a super alloy which is highly chemical resistive used in making digesters, sulfuric reactors etc., by enhancing the tool life and surface finish.

### 3.2.5. Milling on Titanium Alloys

Kim et al. [91] reported enhancement in the micro-milling performance of Ti-6Al-4V alloy with a significant reduction in milling forces, surface roughness, tool wear, and coefficient of friction with simultaneous usage of vegetable oil-based 0.1 wt.% diamond nano fluid under MQL along with $CO_2$ chilly gas at $-25$ °C. The graphene nanoparticles enhanced the lubricating performance of vegetable oil-based cutting fluid in milling TC4 alloy under MQL as studied by Li et al. [92] on sustainable milling of titanium TC4 alloy. The spherical-shaped $Al_2O_3$ nanofluid with MQL followed by $SiO_2$ nanofluid achieved a significant reduction in cutting force and surface roughness with small strips on the debris surface compared to $MoS_2$, CNTs, SiC, and graphite during the investigations by Bai et al. [93] on sustainable milling of Ti-6Al-4V. Osman et al. [94] clarified that 24.75% concentrated ester-based h-BN nano cutting fluid with a 40.115 mL/h flow rate achieves good milling performance with a reduction in milling forces and surface roughness. Jamil et al. [95] claimed that surface roughness, tool wear and energy consumption involved in milling Ti-6Al-4V could be greatly reduced with the mist spraying of Blaso cut oil emulsion-based 1 vol.%, $Al_2O_3$-MWCNT (90:10) nano-cutting fluid under MQL.

### *3.3. Nanofluid-Assisted MQL Drilling*

Many researchers have carried out the drilling operation on various materials as workpieces, and the summarized data is given in Table 3. Further, the section compiles the investigations on drilling operations carried out on different materials.

**Table 3.** Summarized data of drilling operation on different materials using NMQL.

| Ref. No. | Work/Tool Material | Base Fluid | Nano Particles/ Size in nm | Cutting Parameters | | | MQL Parameters | | Effects (Reduction of Responses) |
|---|---|---|---|---|---|---|---|---|---|
| | | | | $V_c$ (m/min) | f (mm/s) | d (mm) | Pressure (bar) | Flow Rate (mL/Min) | |
| [96] | AISI-P20 steel (150 × 150 × 40 mm³)/TiAlN-coated carbide (Ø8.5 mm) | Soluble coconut oil | CuO/<50 | 50–150 | 0.01 | | 6 | 0.166 | 0.5 wt.% con.—tool wear |
| [97] | AISI-4140 steel (130 × 180 × 5 mm³)/carbide drill (Ø10 mm) | ethylene glycol | Cu/70 | 61–123 | 0.02–0.05 | | 4 | 10 | 0.2 wt.% con.—surface roughness and flank wear by 71% and 53% compared to CC and coconut oil cutting environment |
| [98] | AISI-314 stainless steel (30 mm thick)/M35 HSS drill (Ø8 mm) | Sun flower oil | Graphene/ 10 | 7.91 | 0.125 | | 6 | 2 | 1.5 wt.% con.—thrust force, torque, surface roughness, and co-efficient of friction, wear rate compared to pure MQL |
| [99] | Hardox-500 (150 × 100 × 15 mm³)/carbide drill with TiAlCN coating | Rice bran oil and Water-based emulsion | Al₂O₃/ 30 | 15–25 | 0.02–0.06 | | 6 | 0.5 | 1 wt.% con.—surface roughness and thrust force followed by better tool life and surface micro structure of work. |
| [100] | Ti-6Al-4V (30 × 30 × 5 mm³)/Uncoated tungsten carbide (Ø300 µm) | Palm oil | Dimond/ 35 and 80 | 56.52 | 0.0001–0.0008 | 0.4 | 3 | 0.125 | 0.4 wt.% con.—tool wear, torque, and force |
| [101] | Ni-Ti alloy (94 × 70 × 10 mm³)/TiAlN-coated Tungsten carbide drill (Ø6 mm) | sol-cut oil | Al₂O₃/ <50 | 10–30 | 0.02 | - | - | 0.83 | 0.4 wt.% con.—cutting force, surface roughness and tool wear at low cutting speed but failed in achieving the same at high cutting speed |
| [102] | Aluminum 6061 alloy/Uncoated carbide drill of (Ø200 µm) | Paraffin and vegetable oil | Dimond/ 30 | 37.68 | 0.00083 | 0.4 | - | - | Paraffin diamond nano fluid of 1 vol.% con.—torque and thrust force and achieved no burrs compare to other cutting environments |
| [103] | Aluminium-6063 (25 mm thick)/HSS drill (Ø6 mm) | Soya bean oil | Al₂O₃/ 20 | 30–53.7 | 0.037 | - | 4.8 | 3.33 | 1.5 wt.% con.—thrust force, torque, tool wear and surface roughness compared to dry, flood, and pure MQL |
| [104] | Aluminium-6061-T6 alloy plate (20 mm thick)/HSS drill (9, 10 and 11 mm) | Canola oil | MoS₂/ 30 | 28.26–69.08 | 0.02 | - | 10 | 0.5 | 3 wt.% con.—power, force, tool wear and surface roughness compared to other cutting environments. |
| [105] | AA 5052 (150 × 200 × 5 mm³)/Tungsten carbide drill | Ethylene glycol | Cu/ 50 | 60–210 | 150 mm/min | - | 4 | 8 | 0.2 wt.% con.—surface roughness, flank wear and cutting temperature significantly |
| [106] | Compacted Graphite Iron (CGI)/WC-CO twist type drill (Ø6.35 mm) | Misty Blue TM | WS₂ and h-BN/ <100 | 50 | 0.1 | - | 6.8 | 0.2 | 0.1 wt.% con. of WS2 and h-BNnano particles improved the tribological properties by enhancing the lubricating performance of base fluid |

### 3.3.1. Drilling on Steel

Soluble coconut oil (SCO) is a mix of coconut oil with additives such as acacia powder, distilled water, and lime juice. This, along with 0.5 wt.%CuO nano fluid, achieved significant lubricant performance during drilling experiments by Jamil et al. [96] on AISI-P20 steel. The Cu nanofluid with MQL at a cutting speed of 06 m/min and feedrate50 mm/min was found to be the optimum cutting conditions given by Muthuvel et al. [97] with a reduction in surface roughness and flank wear by 71% and 53% respectively with a conventional flood cooling and coconut oil cutting environment. Pal et al. [98] investigated the performance of MQL drilling of AISI-321 stainless steel using graphene nanoparticles and showed that 1.5 wt.% graphene nanofluid with MQL gave the best performance with a reduction in thrust force, torque, surface roughness, and coefficient of friction to 27.4, 64.9, 33.8, and 51.7%, respectively, in comparison to pure MQL. About 4.5 times tool life enhancement was found during hard drilling of difficult-to-cut material, i.e., Hardox-500 steel using the MQCL technique with water emulsion-based $Al_2O_3$ nano cutting fluid by Duc et al. [99].

### 3.3.2. Drilling on Titanium Alloy

Diamond nanofluid (35 nm. 0.4 wt.%) with low F = 10 mm/min achieved a superior reduction in drill torque and thrust force compared to dry and pure MQL during the machinability study of Ti-6Al-4V drilling by Nam and Lee [100]. According to Rosnan et al. [101], usage of sol-cut oil-based $Al_2O_3$ nano-cutting fluid under MQL is not a good option for drilling nickel titanium (Ni-Ti) alloy, a significant material used in orthodontics applications since it failed to achieve low surface roughness, thrust force, dimensional accuracy, and tool wear.

### 3.3.3. Drilling on Aluminum Alloy

Micro drilling on Al6061 alloy with paraffin and vegetable oil diamond nano cutting fluid was performed by Nam et al. [102] and concluded that paraffin diamond nanofluid of 1 vol.% concentration reduces the torque and cutting force and achieves a tool life of 150 high-quality holes. $Al_2O_3$ nanofluid with MQL enhances the drilling performance by reducing the thrust force, torque, and surface roughness with a tool life of 200 holes, whereas dry drilling had shown adhesion at the 27th hole during the study by Chatha et al. [103] in Al6063 alloy. Singh et al. [104] tried using MQL with a Ranque-Hilsch vortex tube (RHVT) as a base lubrication technique. They demonstrated that $MoS_2$ nanofluid with RHVTMQL is significant for sustainable drilling of Al-6061-T6 alloy with an 8% reduction in surface roughness, force, and power compared to dry and canola oil MQL. Babu and Muthukrishnan [105] declared that ethylene glycol-based Cu nano cutting fluid usage in drilling AA-5052alloy enhances the drilling performance with a reduction in surface roughness and flank wear by 92.76% and 36.24% as compared to dry machining and oil MQL.

### 3.3.4. Drilling on Compacted Graphite Iron

Abad and Veldhuis [106] studied the utilization of inorganic nanoparticles namely $WS_2$ and h-BN, as lubricant additives while machining compacted graphite iron (CGI) and found that both WS2 and h-BN nano fluids evenly reduced the coefficient of friction and drilling forces with no significant difference. It was observed that there was no increase in thrust force till the 1950th hole drilling with $WS_2$ and h-BN nanofluids.

### 3.4. Nanofluid-Assisted MQL Grinding

Many researchers performed the grinding operation on various materials as workpieces, and the summarized data is given in Table 4. The section compiles the in-vestigations on the grinding process carried out on different materials.

**Table 4.** Summarized data of grinding operation on different materials using NMQL.

| Ref. No | Work/Tool Material | Base Fluid | Nano Particles/ Size in nm | Cutting Parameters | | | MQL Parameters | | Effects (Reduction of Responses) |
|---------|--------------------|------------|-----------------------------|---------------------|---|---|-----------------|---|-----------------------------------|
| | | | | $V_c$ (m/min) | f (mm/rev) | d (mm) | Pressure (bar) | Flow Rate (mL/Min) | |
| [107] | SK-41C tool steel Width = 2 mm and length = 20 mm/Shank Diameter Φ3 mm, Tool Diameter Φ1.0 mm | Paraffin oil | Diamond/ 30 and 150 | 251.2 | 0.0015 | 0.005 | 3.9 | 0.125 | 2 wt.% con. and 30 nm size—grinding forces and surface roughness compared to other cutting environments |
| [108] | 45 steel/corundum wheel of (300 × 20 × 76.2 mm) | Paraffin, palm oil, rapeseed oil and soya bean oil | $MoS_2$/50 | 1800 | 50 | 0.01 | 6 | 0.833 | 6 wt.% MoS2 con.—co-efficient of friction, specific energy of grinding and surface roughness of the work |
| [109] | AISI-52100 (70 × 50 × 10 mm³)/A60K5V8 wheel with φ200 mm, width: 25 mm Bore (Ø50 mm) | Deionized water | MWCNT/50 | 1500 | 100–166 | 0.01–0.02 | 4 | 0.83–5.8 | 0.81 wt.% con.—best retention of grit sharpness and best dissipation of heat from grinding zone |
| [110] | AISI-52100 (70 × 50 × 10 mm³)/Vitrified alumina wheel | Sunflower oil and synthetic soluble oil | MWCNT/ 40 | 1500 | 83.3–100 | 0.005–0.2 | 3 | 0.83–5.8 | 1 wt.% con.—given more superior cooling and lubrication, wettability and tribological characteristics than all other cutting conditions |
| [111] | AISI-52100 hardened steel (Ø 55 and 15 mm)/Norton e 5SG46-JVS (177 15) | Paraffin (mineral oil) and soya bean (vegetable oil) | $MoS_2$/40 | 2100 | 83.3 | 0.015 | 4 | 5 | UAG with MoS2- grinding forces, G-ratio, and surface quality |
| [112] | AISI-1045 steel (60 × 20 × 10 mm³)/Vitrified Alumina (WA46K5V) (Ø250 mm and 32 mm width) | Canola emulsion | CuO/<50 | 1800 | 83.3–250 | 0.01–0.04 | 4 | 0.83 | 0.15–0.35 vol.% con.—grinding forces, G-ratio, temperature, and surface roughness compared to dry and flood cooling condition |
| [113] | AISI-202 stainless steel (80 × 45 × 8 mm³) | Rapeseed vegetable oil | $MoS_2$/ 50–100 | 3720 | 66 | 0.015 | 4.13–6.2 | 1–2 | 1 wt.% con.—surface roughness, grinding forces and temperature compared to other cutting environments |
| [114] | GH4169 Inconel-718 Ni-based alloy/corundum wheel of (300 × 20 × 76.2 mm) | Synthetic lipids | $MoS_2$ and CNT/ 30 | 1800 | 50 | 0.01 | 6 | 0.83 | 8 wt.%—G-ratio and surface roughness. |

**Table 4.** *Cont.*

| Ref. No | Work/Tool Material | Base Fluid | Nano Particles/ Size in nm | Cutting Parameters | | | MQL Parameters | | Effects (Reduction of Responses) |
|---|---|---|---|---|---|---|---|---|---|
| | | | | $V_c$ (m/min) | f (mm/rev) | d (mm) | Pressure (bar) | Flow Rate (mL/Min) | |
| [115] | GH4169 Inconel-718 Ni-based alloy ($40 \times 30 \times 30$ mm$^3$)/ corundum wheel of ($300 \times 20 \times 76.2$ mm) | Bluebe#LB-1 and plant oil (synthetic lipids) | $Al_2O_3$-SiC/ 30, 50 and 70 | 1800 | 50 | 0.02 | 6 | 0.83 | 30:70 nano particles size ratio attained good surface finish, largest wetting area, high cross section co-efficient and best morphology of abrasive dust compared to other combinations |
| [116] | Inconel-718/Alumina AA80 K5 V8 | Groundnut oil, palm oil | $Al_2O_3$/<100 | 1680 | 35 | 0.02 | 4–6 | 1 | 0.5 wt.% concentrated palm oil-based nano fluid—surface roughness, G-ratio, grinding energy and co-efficient of friction |
| [117] | Ni-Cr alloy/Alumina AA80 K5 V8 (wheel width-13 mm) | Sunflower oil and rice bran oil | CuO/<100 | 1680 | 35 | 0.02 | 4–6 | 1.66 | 0.5 and 1 wt.% con.—surface roughness and grinding energy |
| [118] | Ductile Cast-iron (QT400–18) ($32 \times 12 \times 12$ mm$^3$)/CBN grinding wheel ($300 \times 20 \times 76.2$ mm) | Soya bean oil | CNT/20 | 1800 | 50 | 0.01 | 6 | 0.583 | 2 wt.% con.—surface roughness and achieved highest G-ratio compared to pure MQL flood and dry grinding environments |
| [119] | YG8 Tungsten carbide ($42 \times 8 \times 4$ mm$^3$)/1A1 ($200 \times 15 \times 51$) K120 N D181 C75 | Paraffin oil and sun flower oil | $Al_2O_3$, graphite and $MoS_2$/ <100 | 1800 | 166.6 | 0.02 | 2 | 1.5 | MoS2 in mineral oil—cutting force and energy when compared to all other cutting fluids |
| [120] | SiC reinforced (1,2 and 3 wt.%) Al-matrix composite (20 mm dia. and 300 mm length)/Aluminium oxide grinding wheel | Cashew nut-based vegetable oil | $TiO_2$/ 20 | 565–942 | 62.8 | 0.01–0.03 | - | - | grinding force and temperature compared to pure oils MQL |
| [121] | Carbon fiber-reinforced polymer/Diamond grinding wheel | Palm Oil | CNT/ 50 | 2400 | 30 | 0.02 | 6 | 1 | surface roughness compared to dry and pure MQL |
| [122] | Ti6Al4V-EI ($150 \times 80 \times 20$ mm$^3$)/Cubic boron nitride ($150 \times 12 \times 31.75$ mm) | synthetic fluid, canola oil, soyabean oil and olive oil | $MoS_2$, Graphite and Graphene/ 8 | 1320 | 50 | 0.01 | 5 | 0.83 | Canola -Graphene 1.5 wt.% con.—grinding forces, surface roughness and specific energy greatly compared to other combinations adopted |

### 3.4.1. Grinding on Steel

MQL with diamond nanofluid of 2 wt.% concentration and 30 nm size reduced the grinding forces and surface roughness by 33.2%, 30.3%, and 64% compared to dry machining as investigated by Lee et al. [107] on meso scale grinding of hardSK-41C tool steel. Plain grinding of 45 steel by Zang et al. [108] using different types of MoS2-dispersed vegetable and mineral oils resulted in soya bean-based $MoS_2$ nano-cutting fluid achieving the best lubrication performance with reduced coefficient of friction, specific grinding energy, and surface roughness. 1 vol.% of MWCNT, which can be blended easily in de-ionized water, achieved better lubrication performance under SQL with a significant reduction in surface roughness, G-ratio, and grinding force in both moderate and aggressive grinding modes of AISI-52100 steel by Kumar and Ghosh [109]. The tribological responses of MWCNT nanofluid in the earlier investigations such as contact angle and coefficient of friction as observed by Kumar and Ghosh [110] in their study on surface grinding of AISI-52100 steel with vegetable oil-based MWCNT nanofluid found that vegetable oil with SQCL showed adequate lubrication and tribological performance compared to conventional fluids. Molaie et al. [111] demonstrated that the grinding performance of AISI-52100 hardened steel could be achieved with a superior reduction in grinding force, G-ratio, and surface roughness by employing ultrasonic-assisted grinding process under MQL along with 6 wt.% concentrated paraffin oil-based $MoS_2$ nanofluid compared to soya bean. The minimum grinding force, G-ratio, temperature, and surface roughness as compared to dry and flood cooling conditions were attained by Shabgard et al. [112] during surface grinding AISI-1040 material with canola-based CuO nanofluid. Pal et al. [113], while grinding AISI-202 stainless steel, has concluded that 1 wt.% concentrated MoS2 nanofluid with MQL at 6 bar pressure and 120 mL/h flow rate is the best grinding condition for AISI-1040 steel due to significant reduction of surface roughness, grinding forces, and temperature.

### 3.4.2. Grinding on Nickel Alloys

The optimal lubrication performance of synthetic lipids-based oil was achieved during grinding as studied by Zhang et al. [114] on GH4169Inconel-718 by dispersing 8% mass fraction of $MoS_2$/CNT hybrid nanoparticles in base oil with a significant reduction in G-ratio and surface roughness. Further, Zhang et al. [115] investigated the influence of nanoparticle size on the machining performance of GH4169 Inconel-718. They reported that a higher nanoparticle size ratio of $Al_2O_3$/SiC was effective in achieving a good material removal rate but lacked in surface finish. In contrast, a low nanoparticle size ratio effectively achieved a good surface finish, the largest wetting area, and the best morphology of abrasive dust, cross-correlation surface function curve profile under 30:70 nanoparticles. According to Virdi et al. [116], 0.5 wt.% concentrated palm oil-based $Al_2O_3$ nanofluid with MQL improves the grinding performance of Inconel-718 by reducing the grinding energy and coefficient of friction along with the responses. Furthermore, the sunflower oil-based CuO nanofluid had shown the best lubricating characteristics by significantly reducing the surface roughness and grinding energy in the study by Virdi et al. [117] on grinding Ni-Cr alloy with vegetable oil-based nanofluid.

### 3.4.3. Grinding on Cast Iron

Gao et al. [118] used 2 wt.% vegetable oil-based CNT nanofluid in the up-grinding of ductile cast iron and achieved the best tribological performance with a notable reduction in grinding force, frictional co-efficient, grinding temperature, G-ratio, and surface roughness.

### 3.4.4. Grinding on Ceramics

The efforts on the investigations by Hosseini et al. [119] with nanofluids in grinding hard ceramic, i.e., WC, have shown that mineral oil-based $Al_2O_3$, graphite, and $MoS_2$ nanofluids with MQL enhances the grinding performance of WC with a promising reduction in grinding force, specific energy compared to vegetable oil among all mineral oil $MoS_2$ nanofluids with MQL.

### 3.4.5. Grinding on Composites

The cylindrical grinding lubrication performance of SiC-reinforced Al-matrix composite is prominent in the automobile industry and can be improved significantly by reducing the grinding force and temperature evolved with vegetable oil-based $TiO_2$ nanofluid as carried by Nandakumar et al. [120]. Investigations by Li et al. [121] assessed the surface morphology of carbon fiber-reinforced polymer while transverse grinding using CNT nanofluid and clarified that palm oil-based CNT nanofluid with MQL notably reduces the spectral width, 3D SEM image topography, and 2D fractal dimension of surface roughness profile compared to dry and oil MQL.

Singh et al. [122] proved that the lubrication performance of the canola oil during grinding Ti-6Al-4V-ELI alloy could be increased prominently with 2D-structured graphene nanoparticles which give a notable reduction in surface roughness, grinding forces, and specific cutting energy as compared to MoS2 and 3D graphite nanofluids.

### 3.5. Mechanisms Involved in NMQL Machining

When the nano fluid mist is sprayed on to the tool work interface during the machining, the nano particles in the mist roll between the tool work interfaces. This is termed as rolling mechanism, as illustrated in Figure 2a, in which sliding friction between the tool work is converted to rolling friction, due to which friction co-efficient gets reduced significantly and helps in achieving low cutting forces [93–95].

Figure 2b illustrates the protective film formation while machining with nanofluid under MQL. It is due to the ability of the nanoparticles to get embedded into the furrows of the tool work interface due to the well adsorbate property of nanoparticles, thereby forming a bearing and act as lubricating media for the formation of protective tribo film [93], which helps in reducing the friction and temperature generated during machining [123–125].

The mending mechanism illustrated in Figure 2c is the ability of the nanoparticles present in the mist sprayed while machining under MQL to get deposited onto the friction surface of the work by compensating the loss of mass, thus improving the lubricating performance by reducing the friction which helps in minimizing the cutting temperature and maximizing the tool life and machined surface quality [126].

Due to the high-pressure usage in MQL, the focus is to improve the nanofluid on the tool work interface while machining. It gives the nanoparticles thrust to tackle the nano-size raised debris on the surface of work and makes them remove their hard abrasive nature to achieve a smooth surface finish on the work surface [127–129]. This type of mechanism is called the polishing mechanism illustrated in the Figure 2d.

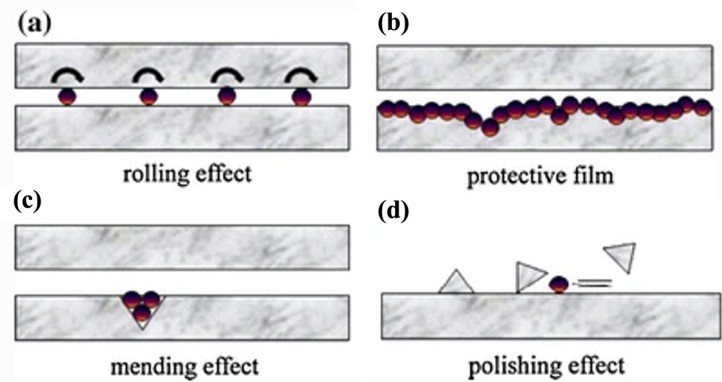

**Figure 2.** Schematic of nanoparticle mechanisms while machining (**a**) rolling, (**b**) protective film, (**c**) mending, and (**d**) polishing [129].

## 4. Cryogenic MQL

González et al. [130] machined Ti6Al4V integral blade rotors (IBR) using $CO_2$ as cryogenic cooling with flank milling technique. This technique uses diamond polycrystalline (PCD) tools using CO2 and minimum quantity lubrication (MQL), denominated CryoMQL,

as an alternative to conventional oil emulsions. The proposed approach implies a balance between technical and environmental issues, and it makes feasible the use of PCD tools to avoid the high temperature reactivity of Ti6Al4V alloy with this type of cutting tools. $CO_2$ must be supplied and injected onto the cutting zone, avoiding the risks of dry ice formation and clogging of both pipes and nozzles. Ihsan et al. [131] developed a new hybrid cryogenic MQL cooling/lubrication technique for end milling Ti-6Al-4V using coated solid carbide tools. The hybrid cryogenic MQL technique shows an increased tool life of 30 times which is achieved together with a 50% improvement in productivity compared to the state-of-the-art flood coolant machining. In their experiments, Jamil et al. [132] obtained the outperformed $CO_2$-snow followed by cryogenic-LN2, MQL, and dry cutting conditions regarding sustainability measures and machining characteristics. The cryogenic-LN2 provided minimum cutting temperature flowed by $CO_2$-snow, MQL, and dry cutting.

Yıldırım et al. [133] observed that the tool wear decreased by 50% in MQL and Cryo-MQL compared with cryogenic machining. Interestingly, they opined that MQL is more effective than cryogenic machining in reducing cutting tool wear. Nagaraj et al. [134] investigated the thrust force, torque, delamination, and diameter error of drilled holes in the drilling of CFRP composites under dry, 67% and 79.60% by using MQL and cryogenic conditions. Better quality of the drilled holes concerning diameter and roundness error was produced under cryogenic conditions because tool life could be improved by maintaining the sharpness of the drill bit since the temperature in the cutting zone is reduced under this condition.

Khanna et al. [135] observed that the performance of sustainable cryogenic machining is superior in terms of lower tool wear, power consumption, and sub-surface microhardness. The finer grain size and helical chip with smaller diameter were produced in the cryogenic machining, while the better surface finish is observed for the flood machining in contrast to other cutting fluid strategies. Chen et al. [136] explored the feasibility of cryogenic minimum quantity lubrication (CMQL) in the precision machining of in situ $TiB_2/7075$ composite, considering the requirements of environmental protection and prolonging tool life. In the supercritical $CO_2$ ($scCO_2$) jet, water-based cutting fluid further improved the cooling performance, while the soluble vegetable oil significantly improved the lubrication performance. The coefficient of friction (COF) under $scCO_2$-oil on water-based MQL ($scCO_2$-WMQL) condition was as low as 0.1 and recorded an increment in tool life by 198.08% compared to dry conditions. Chetan et al. [137] recorded a reduction of 38% in tool flank wear of the 0.2 mm nose radius insert compared to the NMQL technique with the application of liquid nitrogen as a cooling medium. The SEM investigation of the rake face revealed that the cryogenic cooling process has also prevented groove formation and peeling of the coating. The cushions of liquid nitrogen as coolant improved the surface finishing effect compared to other techniques. Yıldırım [138], in his investigation, compared the performance of nano additive-based cutting fluid and cryogenic cooling with liquid nitrogen in terms of machining performance in complex turning processes. He observed that cryogenic cooling was better in tool-chip interface temperature, tool life, tool wear, and chip morphology, whereas nanofluid was better in average surface roughness and surface topography. Kaynak et al. [139] examined the effects of cryogenic cooling on tool-wear rate and progressive tool-wear by comparing the findings from cryogenic machining with results obtained from MQL and dry machining conditions. The results demonstrate that cryogenic cooling has a profound effect on controlling the tool-wear rate and that the progressive tool-wear in the machining of NiTi shape memory alloys can be significantly reduced by cryogenic machining.

## 5. Conclusions

The present work focuses on compiling the previous investigations presenting a comprehensive state-of-the-art with some research gaps on the application of MQL systems in machining different materials. Most of the experimental studies showed that using MQL can be a viable alternative to flood lubrication and facilitate environment-friendly machining. The following conclusions may be drawn from the present literature review:

- Due to the efficient penetration of oil-mist in the contact zone, MQL has shown a significant reduction in the friction coefficient.
- A hybrid cryogenic MQL cooling/lubrication technique for end milling Ti-6Al-4V is developed, which has shown superior performance compared to cryogenic-$LN_2$, MQL, and dry cutting conditions. The cryogenic $LN_2$ showed better performance than MQL in terms of cutting temperature.
- During precision machining of in situ TiB2/7075 composite, the supercritical $CO_2$ ($scCO_2$) jet yielded a coefficient of friction (COF) as low as 0.1 and recorded an increment in tool life by 198.08% compared with dry conditions.
- In the turning process, cryogenic cooling was found better in tool-chip interface temperature, tool life, tool wear, and chip morphology. In contrast, nanofluid showed better results regarding average surface roughness and surface topography.
- Cryogenic cooling has a profound effect on controlling the tool-wear rate and the progressive tool-wear in the machining of NiTi shape memory alloys.
- To get better results from the MQL setup, the designs of the aerosol-supply channels, tool geometry, tool materials, and tool body design (back tapers, reliefs, undercuts and supporting elements) should be optimized for MQL.

Very little work is available on the optimization of MQL operating parameters. Therefore, optimization of MQL operating parameters such as input air pressure, nozzle shape and size, orientation, number of nozzles, and flow of the cutting fluid through the MQL nozzle can be attempted in future. Moreover, the hybridization of the nanoparticles and the cryogenic MQL technique can be explored in the machining of hard-to-machine alloys.

**Author Contributions:** Investigation, A.K.; resources, A.K.; writing—original draft preparation, A.K.S.; writing—review and editing, J.K.K.; supervision, J.K.K.; All authors have read and agreed to the published version of the manuscript.

**Funding:** This research received no external funding.

**Data Availability Statement:** Data sharing is not applicable.

**Conflicts of Interest:** The authors declare no conflict of interest.

## Abbreviations

$V_c$ = cutting speed (m/min)
$f$ = feed (mm/rev)
$d$ = depth of cut (mm)
$L$ = machining length (mm)
$Fz$ = cutting force (N)
$Fy$ = thrust force (N)
$Fx$ = feed force (N)
$VB$ = flank wear of insert (mm)
Eq. = Equations
nps = Nanoparticles
SEM = Scanning electron microscope
XRD = X-ray diffraction
RSM = Response surface methodology
ANOVA = Analysis of variation
$TiO_2$ = Titanium oxide
$Al_2O_3$ = Alumina
MWCNT = Multi-walled carbon nanotubes
$MoS_2$ = Molybdenum disulphide
h-BN = Hexagonal boron nitride
$SiO_2$ = Silicon oxide
$WS_2$ = Tungsten sulfide

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
