# Peer review of "State-of-the-Art in Sustainable Machining of Different Materials Using Nano Minimum Quality Lubrication (NMQL)"

_lubricants, doi:10.3390/lubricants11020064_

Round 1

Reviewer 1 Report

Previous acceptation, authors must be work in some points. Please see the file attached.

Reviewer 2 Report

Dear Authors,

The article titled "State of Art on Sustainable Machining of Different Materials using Nano Minimum Quality Lubrication (NMQL)" can publish in the Journal. However, a radical revision is required for this manuscript. After the major revision, it must be peer-reviewed again.

The article is acceptable after the incorporation of the following significant suggestions:

1- The abstract and introduction need to be reorganized.

2- The nomenclature section can be added before the introduction section to understand the manuscript.

3- The paper's purpose should be presented more strikingly in the introduction section.

4- The whole manuscript was edited according to proofreading typos, such as subscripts and superscripts.

5- A chapter on using surfactants in nanofluid mixtures can be opened. In particular, the table can also give information on the type and use of surfactant.

6- If Table 1 and Table 2 were given horizontally on a separate page, it would be more convenient for the reader.

7- Instead of Figure 2, a different figure describing the mechanisms should be preferred. This is because this figure does not fully express all the nanoparticle mechanisms. In addition, if all nanoparticle mechanisms are added to new figures, the manuscript will be more understandable.

8- The whole manuscript must be checked in terms of English. It should be more understandable for the readers.

9- Please readdressed the conclusion for readers.

Round 2

Reviewer 1 Report

my comments were taken into account. Accepted.

Reviewer 2 Report

The authors addresed reviewer comments, now paper can be publish in the journal.